# Applying Structural Equation Modelling to Understand the Implementation of Social Distancing in the Professional Lives of Healthcare Workers

**DOI:** 10.3390/ijerph20054630

**Published:** 2023-03-06

**Authors:** Muhammad Fayyaz Nazir, Shahzadah Fahed Qureshi

**Affiliations:** 1Department of Public Governance and Management, Faculty of Economics and Business Administration, Ghent University, 9000 Ghent, Belgium; 2Institute of Social and Cultural Studies, Faculty of Arts and Social Sciences, Bahauddin Zakariya University, Multan 60800, Pakistan

**Keywords:** COVID-19, implementation, normalization process theory, nonpharmaceutical interventions, social distancing, structural equation modelling

## Abstract

This study, based on the normalization process theory (NPT), explores the implementation of nonpharmaceutical interventions (NPIs)—specifically social distancing (SD)—in the professional lives of healthcare workers in three Pakistani hospitals. We collected and analysed health workers’ data using partial least squares structural equation modelling (PLS-SEM) and assessed the policy implications of the results. Violations of normality assumptions in the quantitative data and the need for scores of independent variables for follow-up analysis guided the researchers to adopt a structural equation modelling process that involved a stepwise evaluation process for convergent validity, individual item validity, discriminant validity, the structural model relationship, and overall model fitness. Theoretical constructs coherence, cognitive participation, collective action, and reflexive monitoring were found to influence the normalization of SD. The results show that SD was normalized in the professional lives of healthcare workers through strong collective action (resources required) and reflexive monitoring (appraisal) but weak cognitive participation (actors’ engagement) and coherence (sense-making). Low and middle-income countries (LMICs) should work more on actors’ sense-making and engagement in dealing with healthcare crises that require SD. The research findings can aid policy institutions in better understanding the loopholes in the implementation process and making better policies.

## 1. Introduction

This study explores the normalization of social distancing (SD) in the professional lives of healthcare workers. The COVID-19 pandemic has strained global healthcare systems and caused extreme worldwide socioeconomic disruptions, such as the deferment and termination of educational, political, cultural, and religious events [1], ultimately opening a new pandora’s box. The pandemic has motivated policy institutions to provide scientific support. In the absence of pharmaceutical interventions, the World Health Organization (WHO) provides policy guidelines for implementing nonpharmaceutical interventions (NPIs). These include (but are not limited to) contact tracing, quarantine, and SD.

While explaining the importance of SD for low- and middle-income countries (LMICs), Guimaraes et al. [2] demonstrated that other NPIs have greater financial consequences. In Pakistan, the doctor-to-patient ratio in 2019 was 1.1 doctors per one thousand patients. The ratio increased from 0.93 in 2015 and 0.96 in 2016 to 1.0 in 2017 and decreased to 0.98 in 2018 before increasing again in 2019 [3]. However, ratios of these levels are considered to be minimal in terms of the human resources needed to meet human health needs and are below the WHO standards [4]. The lack of human health resources in a country with a large population makes implementing social measures difficult, specifically for healthcare workers in their professional lives. Therefore, it is pertinent for Pakistan to understand the implementation of NPIs and, more specifically, social measures such as SD. Healthcare workers must follow SD because they are more exposed to the disease because of the nature of their work. Razaq et al. [5] observed that 10% of the reported infections were among healthcare workers. If healthcare workers do not follow SD, they could be disease carriers. Considering the importance of SD in the context of healthcare workers’ professional lives, we took SD as a core social measure in this study.

Policy implementation researchers focus on various aspects of policy implementation and explore its context, enablers, challenges, innovation, and multi-actor perspectives [6]. In studying the pandemic, Lotta et al. [7] observed the vital role of frontline workers. To comprehend the implementation context, it is pertinent to involve the policy actors. Anthony [8] concluded that it is important for healthcare workers to observe SD. It is pertinent to explore the normalization of SD in the professional lives of healthcare workers in local contexts so that developing countries can develop and implement better policies in the event of any future pandemic. Our study focused on the following core research question: What is the normalization process of SD in the professional lives of healthcare workers?

The implementation of healthcare interventions can be disrupted in multiple ways: patient examination protocols, the nature of the disease, the patient–doctor relationship, and the broader policy context. Consequently, there is a need to examine how social measures are implemented by policy institutions and adopted by the target population. This study focused on the implementation of a complex healthcare intervention in a focused and micro context, i.e., in the professional lives of healthcare workers. This paper describes how the relevant policy actors (the healthcare workers) made sense of the intervention, how they engaged with it, what resources were required, and how they could appraise the intervention. This study focused on the implementation of a specific social intervention, SD, in clinical settings.

Healthcare intervention is considered a complex intervention. Over the last few decades, many conceptual frameworks, models, and theories have been proposed, tested, and recommended to understand and explore complex interventions [9,10], making the study of healthcare interventions an interesting and challenging domain. Understanding the overall process of implementing SD requires a robust methodological design. For this study, we adopted the normalization process theory (NPT), which is discussed extensively in the implementation literature and in explaining the implementation of complex and innovative healthcare interventions [11]. The implementation of SD in managing COVID-19 is a unique and complex healthcare intervention. NPT provided the theoretical framework for understanding the implementation and normalization of SD in the professional lives of healthcare workers. For the purpose of data collection, we selected one district in Pakistan that was in the red zone regarding COVID-19 infections. Data analysis was performed using the SmartPLS software and following a stepwise process. The study results explain the relationships between the independent variables in the implementation of SD in a focused and micro context, i.e., in the professional lives of healthcare workers.

The complex environment of the COVID-19 pandemic, the importance of SD for LMICS, and the policy actors (healthcare workers) involved provided the impetus for this study. The study contributes to the methodological aspect of the application of NPT and quantitative design and explores the current implementation dilemma of COVID-19. This study addresses translational gaps, including know–do, policy research practice, and think–do. The study was intended to provide guidance to LMICs in successfully implementing complex healthcare interventions in the current and future healthcare crises and better orienting frontline workers and healthcare workers towards implementing healthcare interventions. The study satisfies the current implementation paradigms and contributes to implementation, pandemic, and structural equation modelling (SEM) in the literature.

Section 1.1 explains the need for and objectives of the study. Section 2 presents a review of the literature on the normalization of healthcare interventions. Section 3 and Section 4 present the theoretical model, constructs, and hypotheses tested. Section 5 describes the methods and data used—specifically, the research design, ethical procedures statement, study area, research instrument, sample design, data coding, data collection, testing for a common source bias, ANOVA-based tests (assumptions of normality and non-parametric analysis), and finally, the application of SEM and its reasoning. Section 6 presents the results obtained. Section 7 presents a summary and discussion of the results, the limitations of the study, the contributions of the study, management, and policy implications of the findings, and recommendations for future research. Section 8 presents a concise conclusion summarizing the study methods and key findings.

### 1.1. Need for and Objectives of the Study

Calls from policy implementation researchers (about the need for more studies on theory testing, theory building, and quantitative approaches) and contemporary research trends on various aspects of the COVID-19 pandemic have motivated researchers to explore the implementation of NPIs, specifically SD, as in our case, according to sound theoretical designs. To address this need, we conducted a scientific inquiry with the following objectives:To explore the contemporary dilemma of how to satisfy the calls of policy implementation researchers with sound theoretical support and quantitative research design.To develop guidance for other low and middle-income countries by identifying the weak areas in order to normalize and ultimately implement healthcare intervention as a novel and complex intervention and address translational gaps.

## 2. Previous Research on Healthcare Intervention and SD

For this study, we employed a systematic inquiry of the web of science using the keyword “COVID-19” to search for articles that were published from 2019 to 2021 (10,438) in journals of health policy and public administration (2267), focused on social distancing (141). We excluded 86 articles unrelated to policy perspectives and obtained a list of 55 articles for the full review.

Because the goal of this study was to examine the normalization of SD, we also conducted a search of the web of science database for the literature on normalization using the keyword “normalization.” We found 325 articles published in public administration journals but eliminated 294 that were not related to healthcare intervention, obtaining a list of 31 articles for the full review. Combining the 55 papers on SD with the 31 papers on the normalization of healthcare intervention, excluding duplication, and including additional sources gave us a complete list of 86 publications for a full-text review, as illustrated in Figure 1.

SD refers to increasing space between people to minimize disease spread [12]. SD is an important measure that has worldwide acceptability in managing the COVID-19 pandemic [13]. Researchers, practitioners, and policy institutions have shown interest in exploring various aspects of SD in the context of the pandemic, such as the observance of SD [14,15,16,17]. Some studies have focused on the effects of SD on disease patterns [18], in homeless people [19], and on mental health [20], and access to healthcare facilities [21]. The literature also illustrates the relation of SD to ageism [22], economy [17], healthcare system capacity [23], politics [24], belief systems [25], executive orders of state [26], and the role of the press in implementing SD [27].

A large number of studies were found on various aspects of SD, such as compliance with SD [28,29,30,31,32], the impact of SD [33,34], and the need for SD to protect healthcare workers [8]. However, no study was found on the normalization of SD as a complex healthcare or novel nonpharmaceutical intervention in the professional lives of healthcare workers.

The implementation of the literature explains the normalization of healthcare interventions with various theories, including NPT. This theory helps us understand how any healthcare interventions become part of normal life activities. NPT has four constructs: coherence, cognitive participation, collective action, and reflexive monitoring. The literature illustrates that researchers understand coherence as the extent to which policy actors develop an understanding and the sense of a new practice or intervention [35,36,37,38] and how the new practice or intervention is similar to or different from existing ones [38,39,40]. Cognitive participation can be interpreted as the extent to which policy actors are inclined toward new practice or intervention [41,42], are engaged or involved with the intervention [37,43,44], and are committed to the intervention [35,39]. Collective action is the allocation of individual and collective or organizational resources to the new intervention [39,42], how the intervention is operationalised [45,46], and how the roles and responsibilities of policy actors in implementing the intervention are defined [42]. Reflexive monitoring represents the extent to which the intervention is subject to appraisal [37,46], assessing the impact of the intervention [44], and ensuring the sustainability of change [35].

## 3. Theoretical Model

In the implementation research, one theory that has been discussed broadly is the ‘NPT’, which has been widely promoted to understand policy implementation [11]. Theory framework is given in Figure 2. NPT accentuates differences in the implementation process between the context, actors, and objects, consistent with social and interactive research models [47]. The theory originated from studies that explored the implementation of complex and innovative interventions within healthcare settings [48]. NPT is highly relevant to our research interest, namely, due to the understanding it provides for the implementation of SD as a complex and innovative intervention. The four constructs on which NPT is based—coherence, cognitive participation, collective action, and reflexive monitoring—are discussed below.

The coherence construct represents the planning phase. Sense-making work that is concerned with exploring what people undertake to comprehend a new intervention or practice can be conducted individually or in partnership with other participants. This construct comprises four distinct working processes, i.e., differentiation, communal specification, individual specification, and internalization. For strong coherence, there should be a shared understanding across the policy actors of what this work will entail for individuals. There should be a mutual understanding of the value and purpose of SD.

Cognitive participation is another planning phase construct that is concerned with participation that explains people’s relational activities in reaching a shared agreement for a new practice or intervention. This construct is related to identifying and exploring the specific work that people perform when they try to organize themselves or others to begin a new practice or intervention. Its four working mechanisms are initiation, enrolment, legitimation, and activation. There should be legitimate reasons and strong motivation for policy actors to engage with SD for strong cognitive participation.

Collective action represents the doing phase. It concerns the policy actor’s self-monitoring of their work and exploring what policy actors or people undertake while enacting an intervention or a new practice. This construct comprises four working mechanisms: interactional workability, relational integration, skill-set workability, and contextual integration. For decisive collective action, there should be a shared understanding of the roles and responsibilities of policy actors, and the required resources should be available.

Reflexive monitoring represents the appraisal phase. It is concerned with the evaluation of work that has been performed in the preceding phase. This can be understood as the formal and informal processes that are involved in monitoring and evaluating the work performed in the collective action phase. This stage can be commenced both individually and collectively. This stage comprises four working mechanisms: systematization, communal appraisal, individual appraisal, and reconfiguration. There should be precise evaluation mechanisms for strong reflexive monitoring, and policy actors should be aware of the changes required to adopt SD.

## 4. Hypothesis Development

In this study, coherence is defined as the extent to which healthcare workers understand SD as a healthcare intervention [42]. This involves creating a sense of the intervention through the work that policy actors do individually or collectively to understand the complex intervention. In our case, this illustrates the work of healthcare workers to understand SD. If healthcare workers understand SD, the result is higher coherence, leading to the normalization of SD.

**H1.** 
*Coherence to Normalization of Social Distancing*


Cognitive participation is the extent to which healthcare workers become engaged or involved in implementing SD [37], including engagement through investment in understanding and implementing SD. If healthcare workers are engaged in implementing SD, there is a higher level of cognitive participation, ultimately leading to the normalization of SD.

**H2.** 
*Cognitive Participation to Normalization of Social Distancing*


Collective action is defined as allocating individual and collective or organizational resources to implementing SD [39]. This involves bringing SD into practice through its adaptation of the collective efforts of healthcare workers through the allocation of the resources and development of the skills required to adopt this intervention. If healthcare workers are provided with the organizational or individual resources needed to implement SD, there will be a higher level of collective action, which can lead to the normalization of SD.

**H3.** 
*Collective Action to Normalization of Social Distancing*


Reflexive monitoring is the extent to which SD is subject to appraisal [41]. This involves evaluating interventions by appraising the acceptance of SD among healthcare workers. If there is a higher degree of acceptance for SD among healthcare workers, there will be higher reflexive monitoring, ultimately leading to higher reflexive monitoring.

**H4.** 
*Reflexive Monitoring to Normalization of Social Distancing*


## 5. Methods and Data

### 5.1. Research Design

This study was conducted to investigate the relationship between the pre-defined NPT latent variables and the normalization of SD for the purpose of understanding the process normalization of SD in the professional lives of healthcare workers. Our goal was to explore the weak segments of SD implementation in the local context in Pakistan. We selected latent variables based on their frequency of use and the availability of statistical tools for their assessment [49]. We needed to emphasize the latent variables’ scores for follow-up analysis, so we selected partial least squares structural equation modelling (PLS-SEM) for data analysis, consistent with the recommendation of Hair et al. [50], that researchers should select PLS-SEM when research is based on inquiring about the scores of latent variables for further analysis and identifying the relationships between the dependent and independent variables. Conceptual framework of the study is given in Figure 3. We found PLS-SEM to be suitable for use in this study to quantify the results and draw conclusions. To focus on the local context, we selected one district (Multan) in a highly populated province of Pakistan that was in the red zone regarding COVID-19 infections. In Pakistan, the healthcare structure has three layers. We targeted one hospital from each healthcare domain: Nishter Medical University (NMU) Hospital, a tertiary healthcare facility; District Headquarters (DHQ) Hospital, a primary and secondary healthcare facility; and Fatima Jinnah Women (FJWH) Hospital, a specialized healthcare facility.

### 5.2. Ethical Procedures Statement

Researchers collected data from healthcare professionals. No minors were involved. The data were collected under informed consent protocols and analysed and reported anonymously. Necessary permission from the authors of the NoMAD survey instrument was obtained for its use in our study. Ethical approval was granted by the Faculty of Economics and the Business Administration’s Ethical Review Committee of the first author’s university.

### 5.3. Research Instrument

We used the NoMAD survey instrument, attached as Appendix B, to collect quantitative data. This survey instrument contains four constructs with 23 implementation assessment items or questions on an eight-point Likert scale. NoMAD has been reported to have good face validity, construct validity, and internal consistency and is a highly reliable scale [51]. However, Meier and O’Toole [52] explored a limitation of the survey method in the public administration field, known as common source bias (CSB).

### 5.4. Common Source Bias

CSB is a bias in results that are triggered by variables demonstrating a measurement error because of the common survey method. George and Pandey [53] disputed the existence of CSB and illustrated the response of public administration scholars with four critical arguments. They argued that claims about the effects of CSB might be overstated, endorsed by selective evidence, depend upon the nature of variables, and support the survey data in response to the unsoundness of archival data. We adopted their flow diagram (attached as Figure A1) to assess CSB in our study.

### 5.5. Justification of Research Instrument

George and Pandey [22] presented a flow chart to address the issue of survey selection as a single source. Following their flowchart as a guide, we concluded that the survey method for data collection was suitable for use in this study. This is consistent with George and Pandey’s [22] conclusion, based on work by Podsakoff et al. [54], which states that surveys are suitable data collection methods for variables that are intended to capture individuals’ feelings, perceptions, judgments, and beliefs, as in our case.

### 5.6. Sampling

Three hospitals allowed us to collect data. The purposive/convenience sampling technique was used to collect data from a representative set of healthcare workers, including the managers, medical professionals, and allied healthcare professionals working in the COVID-19 wards of the three hospitals. All of the healthcare workers were initially contacted by email and telephone.

### 5.7. Data Coding and Collection

Before the data collection, we designed a specific coding framework for this research, attached as Table A2. Face-to-face surveys were conducted with 288 healthcare workers in accordance with informed consent protocols. The demographics of the survey respondents are summarised in Table A3. The data used in this research are available from the corresponding author upon reasonable request.

### 5.8. Testing Common Source Bias

To validate the selection of the data collection instrument in light of the recent discussion of CSB in public administration scholarship, we adopted Harman’s single-factor test for CSB. The results (attached as Table A4) indicate values with a 42% variance, which is less than the threshold value of 50% [55]. This confirms the lack of CSB in our case.

### 5.9. Descriptive Statistics and ANOVA-Based Tests

We calculated descriptive statistics for the gender, education, age, location, and designation of the respondents and the normalization of SD. We then conducted ANOVA-based tests concerning the outcome variable. Assumptions of normality were considered and resulted in the selection of non-parametric ANOVA-based tests.

### 5.10. Assumptions of Normality

To test the data distribution for our outcome variable and the process of the normalization of SD in the professional lives of healthcare workers, we conducted Kolmogorov–Smirnov and Shapiro–Wilk normality tests [56]. Both tests assume that a significance value of less than 0.05 represents a violation of normality. We tested the normal distribution of the normalization of SD among all the locations, genders, age groups, education levels, and designations of the survey respondents.

Normality tests for gender, education, organization, and designation resulted in *p*-values of 0.000, indicating non-normality and the need for non-parametric ANOVA-based tests [57]. The results for age subgroups resulted in a *p*-value of 0.000 for the first two sub-groups but 0.272 for the third sub-group. The results (given in Table A5) confirm the non-normality of the data distribution.

### 5.11. Non-Parametric Analysis

Given the non-normality of the data, we applied the non-parametric Mann–Whitney U Test [58] for gender and education and the non-parametric Kruskal–Wills test [59] for age, location, and designation. The Mann–Whitney U tests for gender and education yielded significance values of 0.38 and 0.34, respectively: both less than the threshold value of 0.05. The Kruskal–Wills test for organization/location and designation yielded significance values of 0.030 and 0.001, respectively, but for age, the significance value was 0.672. The results of the tests are given in Table A6.

### 5.12. Application of SEM

The structural equation modelling (SEM) approach was adopted for data analysis in light of the non-normality of the data and the requirement of individual scores for the latent variables for analysis. SEM is a second-generation multivariate technique for data analysis and helps in the evaluation of multifaceted relationships [60,61]. We used the SmartPLS software to provide a visualization of the relationships of the variables. We ran three iterations to analyse the PLS model and compared the results for strong analysis. In the first iteration, we detected an issue of multicollinearity (see Table A7), and after resolving this, we ran the second iteration and found that some items’ values were less than the threshold values. After removing these items, we obtained the final SEM model. The results and models that were obtained in the three iterations are given in Figure A2, Figure A3 and Figure A4.

## 6. Results

The SmartPLS software was used to assess the strength of each latent variable/factor affecting the normalization of SD in the local context. The data analysis was based on descriptive statistics and model assessment followed by a stepwise evaluation process, including (i) the determination of convergent validity and individual item validity, (ii) the determination of discriminant validity, (iii) the development of the structural model relationship, and (iv) assessment of the overall fitness of the model. The stepwise evaluation process yielded values of convergent validity and individual item validity through composite reliability (CR) and average variance extracted (AVE) scores for the three iterations, as shown in Figure 4, Figure 5, Figure 6, Figure 7, Figure 8 and Figure 9. The figures illustrate the values of the manifest and latent variables. The CR or AVE is plotted on the y-axis in each figure. The coherence, cogniti (cognitive participation), collecti (collective action), reflexi (reflexive monitoring), or normaliza (normalization) are plotted on the x-axis.

### 6.1. Individual Item Validity and Convergent Validity

The validity and consistency of the variables were analysed using the measurement model. This model verified the convergent validity primarily by evaluating the factor loadings, CR, and AVE. The CR term engages the standardized loadings of the construct’s indicators/items, so it is a better measure of internal consistency [62]. The values of CR represent the levels at which manifest variables indicate the constructs of the latent variables based on the calculation of the homogeneous outer loadings of the indicators for the constructs. Manifest variables with outer loading values > 0.7 are considered highly suitable [63,64], illustrated with green colours in Figure 4, Figure 5, Figure 6, Figure 7, Figure 8 and Figure 9 However, Siqueira et al. [65] showed that loading values as low as 0.4 should be considered acceptable. If omitting manifest variables with values less than 0.4 increases the value of CR, then it is reasonable to omit them in the second iteration. The results are shown in Figure 4 and Figure 5.

After removing item RM1 to resolve the issue of multicollinearity, we ran the second iteration and found that the values of items CA2, CA3, and CA6 were less than 0.4. We, therefore, tested whether removing these items for the final iteration increased the value of CR. The results, which confirm that the value of the manifest variable’s CR increases (see Table 1), are shown in Figure 6 and Figure 7.

However, item RM2 was deleted for the third (final) iteration to achieve the required threshold value of the CR and AVE values representing the convergent validity of the model [66]. The results are shown in Figure 8 and Figure 9.

The change in the latent variable that was relative to its indicator variable is indicated by the AVE value. Hair et al. [62] and Barclay et al. [67] suggest that the latent variables must extract the least amount of 50% of the variance related to their relative construct indicators, making its value > 0.5.

### 6.2. Discriminant Validity

In the next step, the discriminant validity of the measurement model was evaluated. After evaluating the convergent validity and individual item reliability of the measurement model, the discriminant validity of the latent variables was analysed through the values of cross-loadings.

The discriminant validity indicates the degree to which a particular construct is unique from other constructs [68], based on the rule that ‘items must have a greater value of correlation with the specific latent variable that they are believed to gauge in comparison with all other latent variables presented in the model, specifically Chin’s [69]. The discriminant validity evaluation results are shown in Table 2.

The results specify that the values of item indicators are greater for their constructs than all other constructs. The greater values confirm that the manifest variables in each construct represent the assigned construct, confirming the measurement model’s discriminant validity.

### 6.3. Structural Model Relationship

We used the structural model relationship to evaluate the significance of the individual path and the exploratory power of the model. To analyse the strength of a structural model, Hair et al. [70] suggested considering the values of the path coefficient β (beta) and the coefficient of determination (R squared). However, it is necessary to test the significance of the β value when using a *t*-test. As Kushary [71] recommended, we performed this test with the SmartPLS software through a non-parametric bootstrapping procedure that computes *t*-values using a prespecified number of samples. According to Hair et al. [72], acceptable *t*-values are 1.65 at the 10% significance level, 1.96 at the 5% significance level, and 2.58 at the 1% significance level. We used the SmartPLS software to generate 5000 bootstrapping samples to calculate the t-values, as presented in Table 3, along with the path coefficients.

The structural model illustrated in Figure 10 indicates the relationships between the exogenous and endogenous constructs. The model represents the relationships by allocating components of R-squared and path coefficients [62]. The value of β refers to the effect of an exogenous variable on the endogenous variable. The R-squared value represents the amount of variation described by the structural model [73,74]. According to Cohen et al. [75], a suitable model fit must possess an R-squared value greater than 0.26. On the third iteration, our model yielded an R-squared value of 0.872, indicating that the model explains, to a large degree, the normalization of SD with the latent variables. Athab [76] achieved an R-squared value of 0.844 in another study, and a study on the European customer satisfaction index. Askariazad and Babakhani [77] yielded an R-squared value of 0.706, illustrating that high values of R-squared are certainly possible in some cases.

The next step involved assessing and comparing the latent variable’s path coefficients. All the path coefficients had *t*-values greater than the cut-off values for a significance level of 1%, suggesting that all the path coefficients in the assessed model greatly influenced the normalization of SD in the local context of Pakistan.

### 6.4. Overall Fitness of the Model

We next assessed the goodness of fit (GoF) of the model in terms of the geometric mean of the average R^2^ for all endogenous latent variables and the average communality [28,78]. The primary intent of evaluating the GoF was to examine the PLS model’s performance at both the structural and measurement levels, focusing on the model’s predictive power [78]. Previous studies [79,80] have produced guidelines and GoF threshold values for the global validation of PLS models, with a suggested value of 0.50 as the threshold for commonality diverse effect sizes $R^{2}$.

Epostio and Chin [64] suggested the threshold values of GoF-large (0.36), GoF-medium (0.25), and GoF-small (0.10). The overall fitness of the PLS model was calculated using Equation (1).
(1)$$\mathrm{GoF}=\sqrt{\bar{AVE}\times \bar{R^{2}}}$$

The value of $\bar{R^{2}}$ is 0.871. $\bar{AVE}$ is the average of all AVE values, as shown in Table 3. By using the AVE values of all of the exogenous latent variables in the final iteration, the average obtained is 0.617.
(2)$$\mathrm{GoF}=\sqrt{0.617\times 0.872}$$
GoF = 0.538(3)

Since the GoF value calculated in Equation (3) was greater than the cut-off value of 0.36, the model developed for this research study was considered effective in explaining the relationships between the latent variables and the normalization of SD.

## 7. Discussion

### 7.1. Summary of Results

To achieve the cut-off values of CR and AVE, items CA2, CA3, CA6, and RM2 were omitted from the PLS model in the final iteration. For collective action, the value of CR increased from 0.689 to 0.879, and the value of AVE increased from 0.383 to 0.650. For the case of reflexive monitoring, the value of CR increased from 0.870 to 0.911, and the value of AVE increased from 0.589 to 0.729. For CR, values of 0.813, 0.830, 0.879, and 0.911 were obtained for the latent variables CO, CP, CA, and RM, respectively, and for AVE, values of 0.526, 0.558, 0.650, and 0.729, respectively, were obtained. The higher values of cross-loadings on the latent variables confirm that the variables in each construct signify the consigned construct, confirming the discriminant validity of the model.

All the β values had t-values that were larger than the cut-off value for the 1% significance level, indicating that the path coefficients of the latent variables substantially impacted the normalization of SD. The highest β value of 0.331 for collective action indicates its highest significance in the model. Finally, the overall fitness of the model, as measured by the GoF, was 0.535, higher than the threshold value of 0.36 and indicative of a model with good predictive capability.

### 7.2. Discussion

Our research findings point to a vital aspect of the implementation of SD, namely, its normalization in the professional lives of healthcare workers. These study findings contribute to the existing body of knowledge by providing empirical support for the normalization of SD under the pre-defined theoretical constructs of the normalization process theory through its NoMAD survey instrument [51]. In addition, the study examines the relationships of latent variables for the implementation of SD, ultimately applying the normalization process theory in a novel situation.

Our study contributes methodologically to the literature on PLS-SEM by analysing the relationship of theoretically pre-defined latent NPT variables [81,82] to the normalization of SD. PLS-SEM can model latent variables under the conditions of non-normality for small to medium sample sizes [83], amplifying the clarified variation in endogenous constructs and forecast measures for the latent constructs through multiple regressions [84]. Although previous research has revealed the normalization of several healthcare interventions in various healthcare settings [85], the normalization of SD as a nonpharmaceutical healthcare intervention has not been explored extensively. This study is the first attempt to explain the normalization of SD with empirical support and structural equation modelling.

We conducted tests for common source bias, normality, and multicollinearity under the assumption that the items representing the model should be moderately correlated [86]. Initially, the common source bias was assessed through Harman’s single-factor test, and the results confirmed the absence of CSB. Based on the results of normality tests of the collected data, non-parametric tests were selected for use in the data analysis. We tested the normality of the data for all three locations (MNU HJospital, DHQ Hospital, and FJW Hospital), gender (male, female), age groups (21–30, 31–40, and above 40), education (graduate, post-graduate), and designations (doctors, nurses, and allied health professionals) of the survey respondents with Kolmogorov–Smirnov and Shapiro–Wilk tests. The tests for gender, education, organization, and designation detected non-normality because of the nature of the pandemic and the resources engaged in managing the pandemic. The non-parametric Mann–Whitney U tests for gender and education and the non-parametric Kruskal–Wills tests for organization/location and designation yielded values below the threshold values, illustrate the different representations of the gender groups, education, location, and designations of the respondents that existed alongside the implementation of SD into the local context in Pakistan.

To meet the research objectives, we tested four hypotheses (see results in Table 4). However, we considered the path coefficient values of the latent variables or the theoretical constructs for the significance level among the four constructs. The path coefficient values in order from highest to lowest were for collective action (0.299), reflexive monitoring (0.294), cognitive participation (0.226), and coherence (0.202). The omitted variables for the final iteration represented weak relationships among the latent variables and the implementation of SD. We based our conclusions on the path coefficient values, which were obtained after the omission of the weak variables. The PLS-SEM results support all the hypotheses. We conclude that the latent variables that were identified in this research influenced the normalization of SD.

It is worth mentioning that Alimohamadi et al. [18], Moraes [87], and Battiston [88], also detected a positive relation between SD and the latent variables considered in their studies. The results of our study describe a significant relationship between the latent variables and the implementation of SD. The results of this study also indicate a weak level of coherence, i.e., healthcare workers’ understanding of the value and purpose of SD, and weak cognitive participation, i.e., legitimate reasons and strong motivation of the healthcare workers to engage with SD. However, the results indicate a high level of collective action, i.e., understanding of the role, responsibilities, and resources required for healthcare workers to implement SD, and a high level of reflexive monitoring, i.e., a precise evaluation mechanism and healthcare workers’ awareness of the changes required to adopt SD into their professional lives.

Elf et al. [89] demonstrated a good model fit after deleting three items (CA2, CA3, and RM4) in the second iteration for latent variables that were related to collective action and reflexive monitoring, which is consistent with our results. May et al. [90] found that the latent construct of reflexive monitoring was the least applied theoretical construct in the studies, which is a result inconsistent with our specific case.

### 7.3. Limitations of the Study

Limitations of the study include the following:NPT is a widely discussed theory in the policy implementation of the literature. The theory originated in a developed country, and limited evidence of its applicability in an LMIC is a limitation of this study.The application of NPT in a novel and complex situation such as COVID-19 is another limitation of the study.The study was conducted at the peak of COVID-19 infections, which greatly impacted the data collection activity regarding the number of responses.

### 7.4. Contribution of the Study

The study applied NPT to a specific case through PLS-SEM and explored the implementation and normalization of SD among healthcare workers in the local context in Pakistan. The study addresses the calls of policy implementation researchers about the need for theoretical designs and quantitative work in understanding implementation. The study takes the implementation of SD as a case study to understand the normalization of a complex healthcare intervention in the context of a pandemic. The study contributes to normalization, implementation, PLS-SEM, and the pandemic literature in the following ways.

The study explores a healthcare intervention’s implementation, contributing to the implementation literature.The study explores how SD is normalized in healthcare workers’ professional lives, ultimately contributing to the normalization literature.The study applies NPT in the unique and complex situation of a healthcare crisis, and the results confirm the applicability of NPT. The study thereby validates the utility of the NPT in situations of a novel and complex nature.The study is the first attempt to understand the implementation–normalization of a healthcare intervention (SD) in a healthcare crisis according to a sound theory and multivariate statistical modelling to develop a basis for the policy learning of LMICs. The study results have policy implications which are detailed in the next section.

### 7.5. Implications of the Study

Our study findings have management and policy implications. The study addresses translational gaps: the know–do, policy research practice, and think–do.

Well-known and subtle gaps exist when translating research results into policy and practice [91], such as the know–do gap: the development and implementation of innovative healthcare interventions in practice with the intended users [92], such as, in our case, healthcare workers. Our study identifies loopholes in the implementation process by examining the normalization of an innovative healthcare intervention, addressing the know–do gap.Another well-known gap relates to the use of health research results to inform health policy [93]. The results of our study illustrate the weak coherence (sense-making) and cognitive participation (engagement) on the part of healthcare workers that should inform policy institutions to put more effort into the sense-making and engagement of healthcare workers in Pakistan. This can lead to stronger relationships among the latent variables and the implementation–normalization of a healthcare intervention.However, the think–do gap shows that the groups or individuals involved in the implementation should consider the nature and complexity of the implementation work to better inform their actions. Our study explores the implementation of a complex healthcare intervention through the resources they need to enact it and whether they can appraise it. The resources required and the outcome of the appraisal depend upon the nature and complexity of the intervention. Our study also provides an insight into the think–do gap while understanding the normalization of a complex healthcare intervention in the healthcare workers’ professional roles.

Implementation research suggests that the more significant and rational utilization of theoretical approaches can address translational gaps [94] and develop, enhance, and confirm information about the theories [95]. From a theoretical or methodological perspective, our study contributes to the theory application and quantitative design and explores the contemporary implementation dilemma. We applied NPT to understand the implementation of healthcare interventions, and these results could positively contribute to managing future pandemic/healthcare crises. LMICs can learn from this research, conducted in the local context of Pakistan. Policy institutions can improve their policies, and the target audience can better understand loopholes in the implementation process. The study tested and accepted four hypotheses confirming the applicability of NPT (application of the theory) and SEM (quantitative design) on the examination of CCOVID-19 DP&C policy and, specifically, the normalization of SD.

### 7.6. Future Research

The study explains the normalization of SD in the professional lives of healthcare workers in the local context of Pakistan through PLS-SEM. However, the literature argues that more theoretical [79] and quantitative [96] PLS-SEM-based [70] work is required to understand the role of healthcare workers [8]. The literature also illustrates the importance of contextual factors to policy actors when implementing a policy. Therefore, we recommend the exploration of the role of policy actors’ contextual factors in implementing a healthcare intervention with sound theoretical support, either qualitatively or quantitatively.

## 8. Conclusions

In this study, the PLS-SEM stepwise evaluation process (including an assessment of the convergent validity, individual item validity, discriminant validity, structural model relationship, and overall fitness of the model) yielded significant results that ultimately represent the significant relationships of the pre-defined NPT variables with SD. However, items omitted in the iterative process represented a weak relationship in normalizing SD. The study accepted the four hypotheses that were posed after the critical evaluation. The results indicate that SD normalizes healthcare workers’ professional lives through coherence, cognitive participation, collective action, and reflexive monitoring. Collective action had the greatest influence on the normalization of SD in the local context of Multan, Pakistan. Coherence had the least impact. SD was normalized in the professional lives of healthcare workers through strong collective action and reflexive monitoring but was weak in cognitive participation and coherence.

## Figures and Tables

**Figure 1 ijerph-20-04630-f001:**
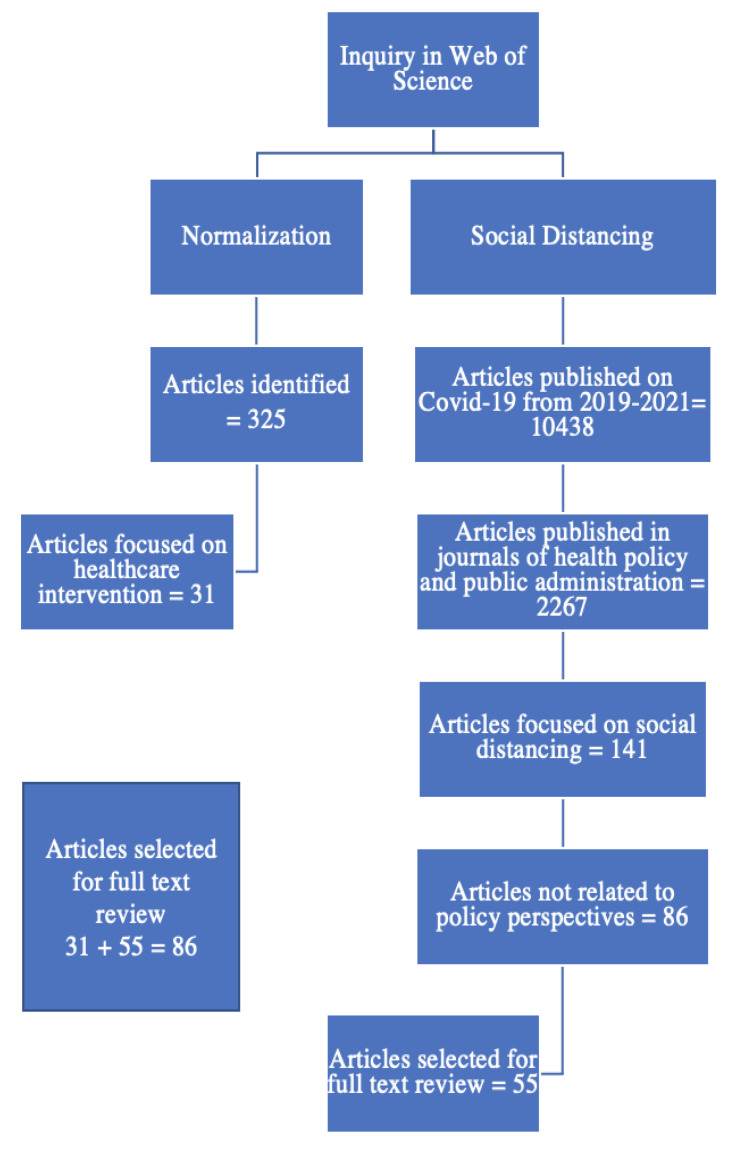
Search strategy of the literature.

**Figure 2 ijerph-20-04630-f002:**
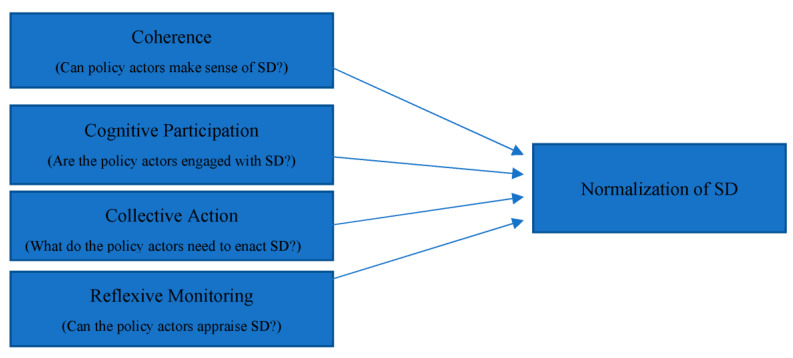
Theoretical framework.

**Figure 3 ijerph-20-04630-f003:**
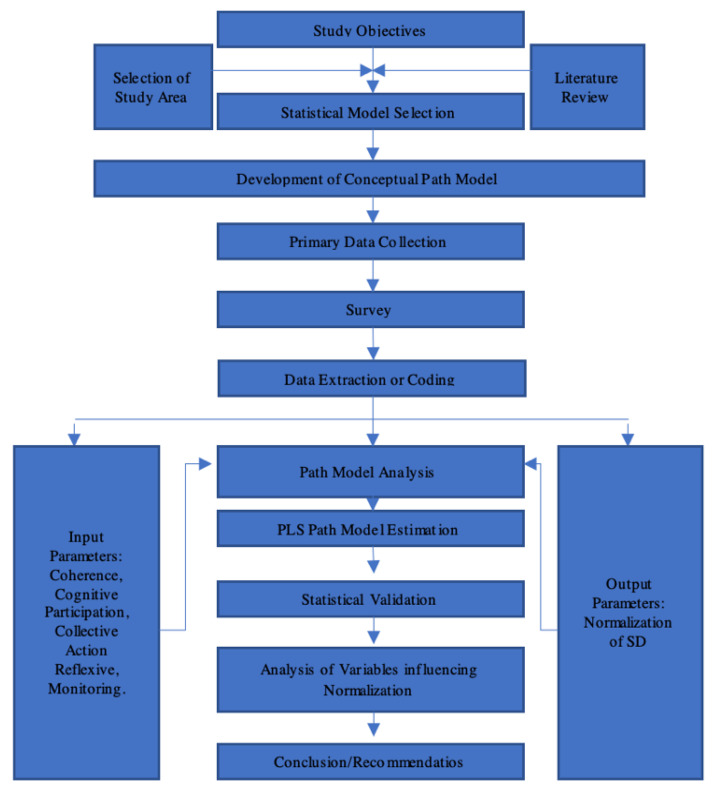
Conceptual framework.

**Figure 4 ijerph-20-04630-f004:**
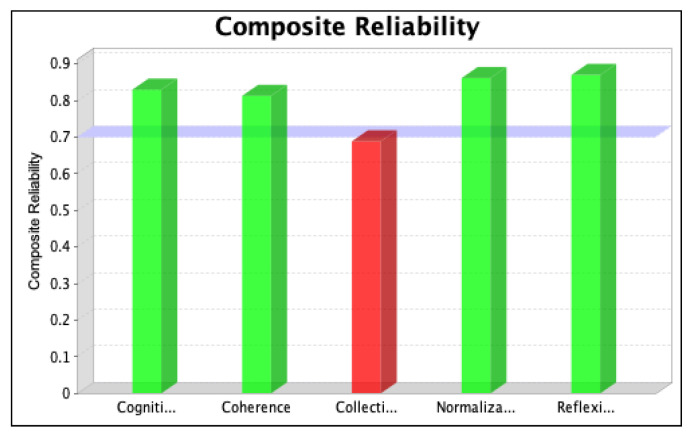
Values of composite reliability in the first iteration.

**Figure 5 ijerph-20-04630-f005:**
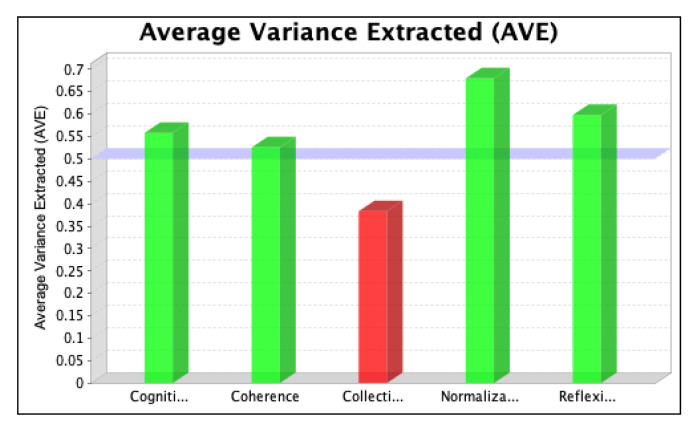
Values of average variance extracted in the first iteration.

**Figure 6 ijerph-20-04630-f006:**
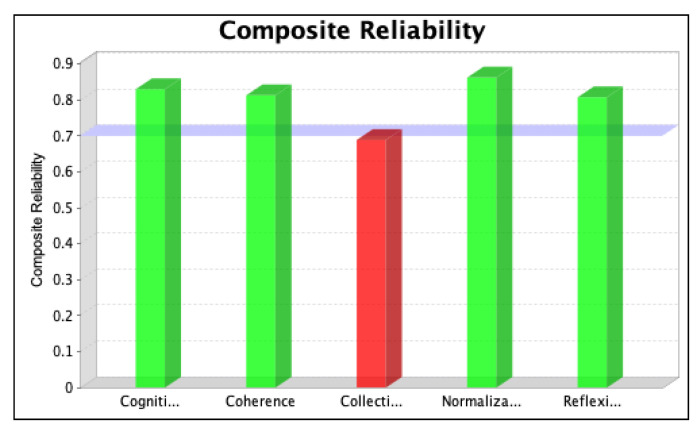
Values of composite reliability in the second iteration.

**Figure 7 ijerph-20-04630-f007:**
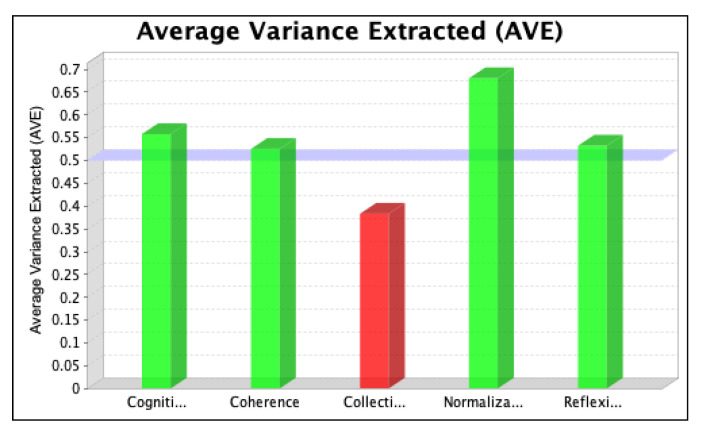
Values of average variance extracted in the second iteration.

**Figure 8 ijerph-20-04630-f008:**
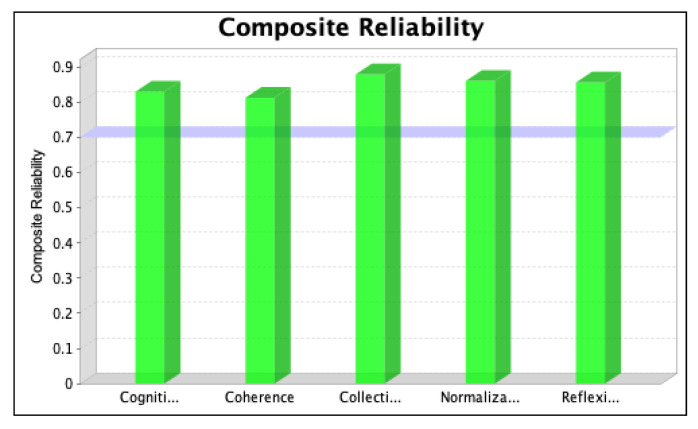
Values of composite reliability in the third iteration.

**Figure 9 ijerph-20-04630-f009:**
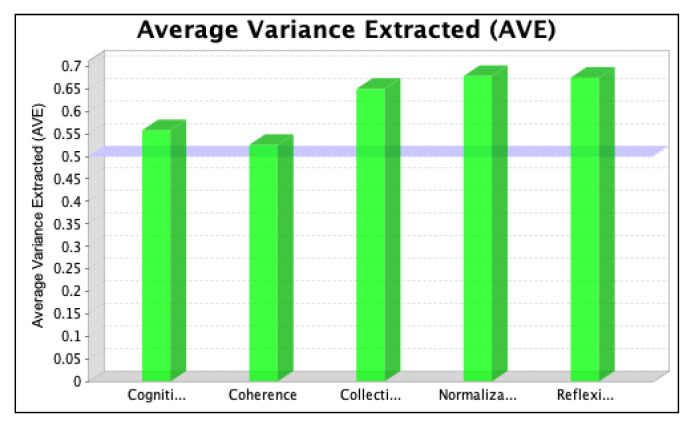
Values of average variance extracted in the third iteration.

**Figure 10 ijerph-20-04630-f010:**
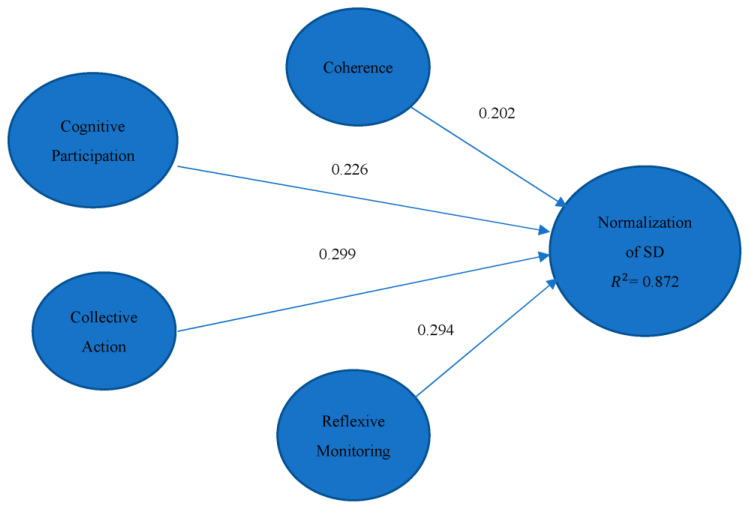
Structural model relationship.

**Table 1 ijerph-20-04630-t001:** Individual item validity and convergent validity.

Constructs	Items	First Iteration			Second Iteration			Third Iteration		
		Loading	CR	AVE	Loading	CR	AVE	Loading	CR	AVE
SD Normalization	NP1 NP2 NP3	0.911 0.897 0.635	0.861	0.680	0.911 0.894 0.642	0.862	0.680	0.915 0.898 0.626	0.860	0.678
Coherence	CO1 CO2 CO3 CO4	0.614 0.684 0.679 0.892	0.813	0.526	0.614 0.684 0.679 0.892	0.813	0526	0.615 0.684 0.680 0.892	0.813	0.526
Cognitive Participation	CP1 CP2 CP3 CP4	0.851 0.496 0.764 0.824	0.830	0.558	0.851 0.496 0.794 0.823	0.830	0.558	0.851 0.496 0.764 0.824	0.830	0558
Collective Action	CA1 CA2 CA3 CA4 CA5 CA6 CA7	0.610 -0.319 0.143 0.859 0.806 0.102 0.890	0.689	0.383	0.611 **−0.320** **0.143** 0.859 0.806 **0.102** 0.890	0.689	**0.383**	0.602 **Omitted** **Omitted** 0.878 0.802 **Omitted** 0.907	0.879	0.650
Reflexive Monitoring	RM1 RM2 RM3 RM4 RM5	**0.942** 0.379 0.484 **0.933** 0.925	0.870	0.598	**Omitted** 0.449 0.537 0.896 0.916	0.807	0.533	- **Omitted** 0.556 0.925 0.927	0.856	0.675

**Table 2 ijerph-20-04630-t002:** Analysis of discriminant validity.

Analysis of Discriminant Validity through Cross-Loadings					
Constructs	Normalization of Social Distancing	Coherence	Cognitive Participation	Collective Action	Reflexive Monitoring
NP1	0.915	0.835	0.813	0.837	0.759
NP2	0.898	0.598	0.842	0.813	0.830
NP3	0.626	0.316	0.409	0.429	0.367
CO1	0.349	0.615	0.267	0.263	0.283
CO2	0.347	0.684	0.296	0.250	0.347
CO3	0.367	0.680	0.326	0.338	0.348
CO4	0.846	0.892	0.809	0.775	0.738
CP1	0.790	0.634	0.851	0.808	0.717
CP2	0.318	0.212	0.496	0.360	0.345
CP3	0.659	0.491	0.764	0.601	0.558
CP4	0.734	0.600	0.824	0.691	0.814
CA1	0.464	0.401	0.440	0.602	0.419
CA4	0.679	0.524	0.701	0.878	0.670
CA5	0.738	0.527	0.784	0.802	0.790
CA7	0.771	0.642	0.756	0.907	0.698
RM3	0.496	0.508	0.393	0.475	0.556
RM4	0.790	0.590	0.828	0.730	0.925
RM5	0.841	0.511	0.837	0.862	0.927

**Table 3 ijerph-20-04630-t003:** Structural model significance.

Constructs	Path Coefficients	Sample Mean	Standard Deviation	t-Value	*p*-Values	Result
CO	0.202	0.201	0.060	3.359	0.000	Significant
CP	0.226	0.233	0.060	3.772	0.001	Significant
CA	0.299	0.297	0.091	3.289	0.001	Significant
RM	0.294	0.290	0.068	4.346	0.000	Significant

**Table 4 ijerph-20-04630-t004:** Descriptive statistics and hypothesis testing.

	Original Sample (O)	Sample Mean (M)	Standard Deviation	T Statistics	*p* Values	Hypothesis Testing
NP1—Normalization of SD	0.915	0.915	0.009	105.203	0.000	
NP2—Normalization of SD	0.898	0.898	0.012	77.241	0.000	
NP3—Normalization of SD	0.626	0.622	0.045	14.021	0.000	
CO1—Coherence	0.615	0.616	0.053	11.657	0.000	
CO2—Coherence	0.684	0.687	0.047	14.509	0.000	
CO3—Coherence	0.680	0.678	0.053	12.932	0.000	
CO4—Coherence	0.892	0.893	0.008	116.133	0.000	
Coherence to Normalization of SD						Supported
CP1—Cognitive Participation	0.851	0.851	0.017	50.169	0.000	
CP2—Cognitive Participation	0.496	0.498	0.097	5.094	0.000	
CP3—Cognitive Participation	0.764	0.762	0.035	22.094	0.000	
CP4—Cognitive Participation	0.824	0.823	0.031	26.958	0.000	
Cognitive Participation to Normalization of SD						Supported
CA1—Collective Action	0.602	0.604	0.060	10.097	0.000	
CA4—Collective Action	0.878	0.878	0.026	33.466	0.000	
CA5—Collective Action	0.802	0.803	0.024	33.711	0.000	
CA7—Collective Action	0.907	0.904	0.021	42.345	0.000	
Collective Action to Normalization of SD						Supported
RM3—Reflexive Monitoring	0.556	0.556	0.064	8.636	0.000	
RM4—Reflexive Monitoring	0.925	0.925	0.019	49.739	0.000	
RM5—Reflexive Monitoring	0.927	0.928	0.010	92.582	0.000	
Reflexive Monitoring to Normalization of SD						Supported

## Data Availability

It is pertinent to mention that the survey data underlying this research will be shared on rational demand with the corresponding author. However, for the reviewers, the data, including the updated/amended survey instrument, coding framework, and PLS-SEM model results, are available at the link: https://drive.google.com/drive/folders/1mbK8y_RZX3WvMxfJL6-xZ6tJKbhzpBBa?usp=sharing. It is declared that the authors obtained permission to use the NoMAD survey instrument tool from its original authors and this can be provided on demand.

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
