# Peer review of "Applying Structural Equation Modelling to Understand the Implementation of Social Distancing in the Professional Lives of Healthcare Workers"

_ijerph, 2023, doi:10.3390/ijerph20054630_

Round 1

Reviewer 1 Report

General Comments:

The authors explore the process of normalization of Social Distancing (S.D.) in the professional lives of healthcare workers. The authors adopted the Normalization Process Theory (NPT) and selected one district in Pakistan that was in the red zone regarding Covid-19 infections to collect data. Data analysis was performed through the Smart-PLS software by following a step-wise process. By explore the effect of four constructs (e.g., coherence, cognitive participation, collective action, and reflexive monitoring) on the normalization of S.D. They find that weak coherence and cognitive participation in the normalization of S.D. They further conclude that the work to improve coherence and cognitive participation may result in stronger relationships among the latent variables and the normalization of S.D.

However, the analysis does not reach a satisfactory level. I do have several major concerns as below.

Major Concerns

1. In the Introduction, the practical and theoretical motivation for Social Distancing is weak.  From the practice, the authors just describe the S.D. is a part of Nonpharmaceutical Interventions in line 39; from the theory, authors refer to Normalization Process Theory but without any explanation, which is not convincing.  In addition, there's no innovation point description and more specific management implication.

2. The motivation of the research is weak. According to the Introduction, “implementing Nonpharmaceutical Interventions (NPIs) includes (but are not limited to) compulsory stay at home, mandatory face masks in public places, and observing S.D.” However, the authors only select S.D. as a healthcare intervention. This is questionable and lacks a theoretical basis.

3. The introduction part lacks a description of innovation and research value, so I think the motivation of this problem is very weak.

4. The literature review structure is disordered; thus, readers don't know the differences between this paper and previous research. What's the innovation of this paper?

5. The literature review focus on Normalizing Healthcare Intervention, but the authors just describe the S.D. in the second paragraph. This is questionable and a clear description is needed.

6. The authors just state, “The theory has four constructs: coherence, cognitive participation, collective action, and reflexive monitoring. Authors observed variations in defining and understanding the above listed-theoretical constructs for their specific use in the normalization literature.” This is confusing and needs a clear description. What’s more, I can’t understand the theory has these four constructs by reading the literature review.

7. In part 4 Materials and Methods, the sample size should be stated.

8. Figure 8 is incomplete. There is no complete presentation of .

9. The results are not convincing enough. The discussion part is only compared with the literature, lacking analysis of the practical application. In addition, the discussion part should analyze the research results and propose management implications, rather than focusing on the data analysis and the contribution to the literature on PLS-SEM, you do not cover this point in your discussion part.

10. The format of the references is inconsistent, and some of the references have the wrong format.

11. The language of the whole article can't meet the requirements of publication, and some expression is difficult for the reader to understand.

Author Response

To

The Editors

International Journal of Environmental Research & Public Health (IJER&PH)

Subject: Reviewer Response Letter

Manuscript ID: ijerph-2099190

Manuscript Title: Applying Structural Equation Modelling for Understanding the Implementation of Social Distancing in the Professional Lives of Healthcare Workers

Dear Editor,

Dear Reviewers,

We hope this finds you well. First of all, please accept our thanks for providing the opportunity to revise the manuscript for consideration in the International Journal of Environmental Research and Public Health. We would also like to thank the reviewers for their constructive comments, which have helped us strengthen and refine this manuscript.

After thoroughly considering each reviewer's comment and undertaking the suggested changes, we are glad to submit the revised manuscript for your consideration. The below response letter offers a detailed reflection on the reviewers' comments and highlights the critical changes undertaken in this revision to address the comments provided.

We hope these changes address the reviewers' comments and remain fully available to provide any additional clarifications or undertake further revisions if required.

Sincerely,

The authors

Reply to Reviewer 1

No.

Comment

Answer/Justification/amendment

1

In the Introduction, the practical and theoretical motivation for Social Distancing is weak.  From the practice, the authors just describe S.D. as a part of Nonpharmaceutical Interventions in line 39;

From the theory, authors refer to Normalization Process Theory without any explanation, which is not convincing.

In addition, there's no innovation point description and more specific management implications.

While explaining the importance of S.D. for Low-Middle Income Countries (LMICs), Guimaraes et al. (2021) demonstrated that other NPIs would increase the financial consequences. So, it is pertinent for LMICs to understand the implementation of S.D.

We enriched the manuscript with a complete subpart: 3-Theoretical Model, where we explain the importance of the NPT in the implementation literature, why we select the NPT, the theoretical constructs, and the theoretical framework.

The complex environment of the Covid-19 pandemic, the importance of S.D. for LMICS, and the policy actors/healthcare workers emanate the need for the study. The study results help the LMICs implement complex healthcare interventions in current and future healthcare crises. The study satisfies the current implementation paradigms and will intensify the implementation, pandemic, and Structural Equation Modelling (SEM) literature

2

The motivation of the research is weak. According to the Introduction, "implementing Nonpharmaceutical Interventions (NPIs) includes (but are not limited to) compulsory stay at home, mandatory face masks in public places, and observing S.D." However, the authors only select S.D. as a healthcare intervention. This is questionable and lacks a theoretical basis.

Authors select S.D. because of the below given points:

1.     Other NPIs would enhance the financial consequences for the LMICs,

2.     It is difficult to follow SD by the healthcare workers because of the nature of their job.

3.     It is pertinent for healthcare workers to observe SD in their professional lives, because they have direct contact with the patients, and they can be the source of disease spread.

3

The introduction lacks a description of innovation and research value, so I think the motivation of this problem is very weak.

The complex environment of the Covid-19 pandemic, the importance of S.D. for LMICS, and the policy actors/healthcare workers emanate the need for the study. The study results help the LMICs to successfully implement complex healthcare interventions in the current and future healthcare crises and can make the frontline workers/healthcare workers more comfortable. The study satisfies the current implementation paradigms and will intensify the implementation, pandemic, and Structural Equation Modelling (SEM) literature.

4

The literature review structure is disordered; thus, readers don't know the differences between this paper and previous research. What's the innovation of this paper?

In this paper, the authors explore the implementation of S.D. in the professional lives of healthcare workers. Razaq et al. (2020) observed that approximately 10% of reported infections were among healthcare workers. Healthcare workers need physical contact with patients, so exploring the implementation of S.D. in the professional lives of healthcare workers is pertinent.

5

The literature review focuses on Normalizing Healthcare Intervention, but the authors just describe S.D. in the second paragraph. This is questionable, and a clear description is needed.

We take S.D. as a complex healthcare intervention, so we explored S.D. and the normalization of a healthcare intervention, merged them into one stream, and developed the normalization of SD.

6

The authors just state, "The theory has four constructs: coherence, cognitive participation, collective action, and reflexive monitoring. Authors observed variations in defining and understanding the above listed-theoretical constructs for their specific use in the normalization literature."

This is confusing and needs a clear description. What's more, I can't understand the theory has these four constructs by reading the literature review

The detail of the four theoretical constructs is given in annex 1.1. However, I shift the necessary detail to the literature part, as given below

The implementation literature explains the normalization of healthcare interventions with various theories, including the NPT. Theory helps us understand how any healthcare intervention became part of normal life activities. The theory has four constructs: coherence, cognitive participation, collective action, and reflexive monitoring. The literature illustrates that researchers understand coherence as the extent to which the policy actors made an understanding and sense of new practice or intervention (Buckingham et al., 2015: Stevenson, 2015: Carter et al., 2016: Porter et al., 2016) and to comprise, how the information of new practice or intervention similar and different from the existing one (Bayliss et al., 2016: Porter et al., 2016, Ricketts et al., 2016). The cognitive participation could be interpreted as the extent to which the policy actors fetched towards the new practice or the intervention (Bouamrane and Mair, 2013: Blickem et al., 2014), engaged or involved with the particular intervention (Carter et al., 2016: Reeve et al., 2018: Teunissen et al., 2017) and are committed with the specific intervention (Buckingham et al., 2015: Stevenson, 2015: Bayliss et al., 2016). The collective action is the allocation of individual and collective or organizational resources to the new intervention (Bayliss et al., 2016, Bouamrane and Mair, 2014), how the intervention was operationalized (Bayliss et al., 2016, Blickem et al., 2014: Bouamrane and Mair, 2013: Buckingham et al., 2015: Coupe et al., 2014: Teunissen et al., 2017) and defining the roles and responsibilities of policy actors (Bouamrane and Mair, 2013). Finally, the reflexive monitoring represents the extent to which the specific intervention is subject to appraisal (Blickem et al., 2014: Coupe et al., 2014: Buckingham et al., 2015: Bayliss et al., 2016: Carter et al., 2016: Teunissen et al., 2017), impact assessment of intervention (Bouamrane and Mair, 2014, Teunissen, et al., 2017) and ensuring the sustainability of change (Buckingham et al., 2015: Carter et al., 2016: Teunissen et al., 2017).

7

In part 4 Materials and Methods, the sample size should be stated

The sample size is mentioned in 4.7 as: Before the data collection. Scholars designed a specific coding framework for this research, attached as annex-2.2. The face-to-face survey was conducted with two hundred and eighty-eight healthcare workers with the informed consent protocols. The description of the demographics of the survey respondents is given in annex-2.3. It is pertinent to mention that the data underlying this research will be shared on rational demand with the corresponding author

8

Figure 8 is incomplete. There is no complete presentation of

Figure replaced. The structural model given in figure 9 indicates the relationship between the exogenous and endogenous constructs. However, the detailed structural models are presented in annex-4.1, 4.2, and 4.3.

9

The results are not convincing enough. The discussion part is only compared with the literature, lacking analysis of the practical application. In addition, the discussion part should analyze the research results and propose management implications rather than focusing on the data analysis and the contribution to the literature on PLS-SEM, you do not cover this point in your discussion part

The updated discussion part is divided into the summary of results, discussion, limitation, contribution, implication, and future research.

10

The format of the references is inconsistent, and some of the references have the wrong format

Thanks for the comment. The issue is resolved

11

The language of the whole article can't meet the requirements of publication, and some expression is difficult for the reader to understand

Thanks for the comment. The issue is resolved

Reviewer 2 Report

The author must include a conceptual model, including hypotheses detail in the figure.

Please explain how the author conducted pilot testing and checked the initial reliability of the questionnaire.

Why did the author use an eight-point Likert scale?

The author must explain the Common Source Bias section in detail and include a minimum of three steps to reduce Common Source Biasness.

The author must justify their sample size.

Figure 8. Structural Model Relationship is not visible.

More studies on methodology are required to support statistical results.

The contribution of the Study section needs to be explained in detail.

Implications of the Study section need to be explained in detail.

The author must include a limitation section in the manuscript.

Support your results with previous research studies based on similarities and dissimilarities.  

To support your results with a few similar studies in the same methodology. You can refer to the provided paper.

Hair, J.F., Risher, J.J., Sarstedt, M. and Ringle, C.M. (2019), "When to use and how to report the results of PLS-SEM", European Business Review, Vol. 31 No. 1, pp. 2-24. https://doi.org/10.1108/EBR-11-2018-0203

Hair, J. F., Ringle, C. M., & Sarstedt, M. (2011). PLS-SEM: Indeed a silver bullet. Journal of Marketing theory and Practice19(2), 139-152.

Author Response

To

The Editors

International Journal of Environmental Research & Public Health (IJER&PH)

Subject: Reviewer Response Letter

Manuscript ID: ijerph-2099190

Manuscript Title: Applying Structural Equation Modelling for Understanding the Implementation of Social Distancing in the Professional Lives of Healthcare Workers

Dear Editor,

Dear Reviewers,

We hope this finds you well. First of all, please accept our thanks for providing the opportunity to revise the manuscript for consideration in the International Journal of Environmental Research and Public Health. We would also like to pay our heartiest thanks to the reviewers for their constructive comments, which have helped us strengthen and refine this manuscript.

After thoroughly considering each comment made by the reviewers and undertaking the suggested changes, we are glad to submit the revised manuscript for your consideration. The below response letter offers a detailed reflection on the reviewers' comments and highlights the key changes undertaken in this revision to address the comments provided.

We hope these changes address the reviewers' comments, and we remain fully available to provide any additional clarifications or undertake any further revisions if required.

Sincerely,

The authors

Reply to Reviewer-2

1

The author must include a conceptual model, including hypotheses detail in the figure

the conceptual model is given in figure 3 on page 6.

We mention the description of the theoretical constructs that developed the basis of the hypothesis in a new sub-part of the study, that is, a theoretical model, on page 4.

2

Please explain how the author conducted pilot testing and checked the initial reliability of the questionnaire

For pilot testing of the instrument, we performed the below-stated process. We analyzed the scale validity of the survey instrument; we calculated the Cronbach alpha value through the Confirmatory Factor Analysis approach. We analyzed the instrument's strength through a step-wise evaluation process, including convergent validity & individual item validity (Values of rho_A), discriminant validity (through Fornell-Larcker Criterion), and model significance (through P-Values).  The initial factor analysis model showed a good fit for two constructs (Coherence and Cognitive Participation) and an unsatisfactory fit for the remaining construct (Collective Action and Reflexive Monitoring). Deleting some items from the collective action and reflexive monitoring, the model yielded a good fit and internal consistency.

3

Why did the author use an eight-point Likert scale?

We adopted the NoMAD with permission from its authors. The actual instrument had an eight-point Likert scale, so we followed the same scale.

4

The author must explain the Common Source Bias section in detail and include a minimum of three steps to reduce Common Source Biasness

We followed the George George et al., 2017 flow chart to justify the selection of the survey tool as a single source for this study. We also conducted the Harmans single factor test (Jung, 2013) to analyze the common source bias (results attached as annex.3.2). The results illustrate the values with 42% variance, less than the threshold value of 50% (Kock, 2020), confirming the nonexistence of common source bias in our case. So, the nonexistence of CSB allows us to further extend the study.

We illustrated the common source bias in sections 5.3, 5.4, 5.5, and 5.8.

5

The author must justify their sample size

The face-to-face survey was conducted with two hundred and eighty-eight healthcare workers with the informed consent protocols. Because of the smaller number of responses, the abnormality of data and the need for the scores of the latent variables guide us to select the structural equation modeling for data analysis. The data collection activity was performed at the peak time of Covid-19 that affected the data collection. This is one of the study's limitations.

6

Figure 8 The structural Model Relationship is not visible

Figure re-designed.

However, The structural model given in figure 9 indicates the relationship between the exogenous and endogenous constructs. The model represents the relationship by allocating the values of the coefficient of determination and path coefficients. (Hair et al., 2012). The coefficient of determination is known as R-square, and the path coefficient is beta (b). Value of b refers to the effect of an exogenous variable on the endogenous variable. The R-square value represents the variance level that could be described by the structural model (Aibinu and Lawati, 2010: Akter et al., 2011). As per Cohen et al. (2013), a suitable model fit must possess an R-square value greater than 0.26. It is worth mentioning that our model on the third iteration gave the value of the R-square 0.872, which is overhead at the suggested level, representing that the developed model is expected to hold a significant degree of explained variance of normalization of SD with the latent variables. The study conducted by Athab (2019) resulted in the value of the R-square as 0.844, and a study on European Customer Satisfaction Index (Askariazad et al., 2015) resulted in an R-square of 0.706, illustrating that the high value of R-square is certainly possible in some cases

7

More studies on methodology are required to support statistical results

We include some more studies based on the same methodology (NPT and SEM) in the introduction, literature, and discussion part of the article.

8

The contribution of the Study section needs to be explained in detail

The study applies the NPT in our specific case through the PLS-SEM and explores the implementation of SD in the local context. So, the study contributes to the normalization, implementation, PLS-SEM literature, and policy learning.

The study explores the implementation of a healthcare intervention, so it contributes to the implementation literature.

The study explores how the SD is normalizing in the professional lives of healthcare workers so, ultimately contributing to the normalizing literature.

The study applies the NPT in a unique and complex situation of a healthcare crisis, and the results confirm the applicability of NPT. So study validates the utility of the NPT in situations of novel and complex nature.

The study is the first attempt to understand the implementation of a healthcare intervention (SD) in a healthcare crisis under a sound theory and multivariate statistical modeling to develop the grounds for policy learning for LMICs. The study also provides the policy implications given in the next section.

9

Implications of the Study section need to be explained in detail

Implications of the study are:

The study results represent weak coherence (healthcare workers' understanding of the SD). So, the work to improve the healthcare workers understanding regarding healthcare intervention would improve the level of implementation.

The study resulted in weak cognitive participation in the normalization of SD. So, the work to improve cognitive participation, such as the healthcare workers' engagement with the SD, would result in stronger relationships among the latent variables and the normalization of SD in LMICs.

Resultantly, this increases the likelihood of the successful implementation of NPIs. Specifically, the study results will positively contribute to managing any future pandemic/healthcare crisis. So, LMICs can learn from this research conducted in the local context of Pakistan. Resultantly, policy institutions can make better policies, and the target audience can better understand the loopholes in the implementation process.

10

The author must include a limitation section in the manuscript

The limitations section is already given in 7.2. However, we improve the section for further clarity and understanding.

Limitations of the study include:

NPT is a widely discussed theory in the policy implementation literature. The theory originated in a developed country, and its application to an LMIC is in itself a limitation of this study.

The application of the NPT in a novel and complex situation of Covid-19 is another limitation of the study.

The study was conducted at the peak of Covid-19 infections, which highly impacted the data collection activity

11

Support your results with previous research studies based on similarities and dissimilarities

The case of Elf et al. (2018) showed a good fit to the model after deleting three items (CA2, CA3, and RM4) for the second iteration from latent variables collective action and reflexive monitoring, making it consistent with our study. Additionally, May et al. (2018) revealed that the latent construct reflexive monitoring is the least applied theoretical construct in the studies, giving us one inconsistent result with our specific case

12

To support your results with a few similar studies in the same methodology. You can refer to the provided paper

Hair, J.F., Risher, J.J., Sarstedt, M. and Ringle, C.M. (2019), "When to use and how to report the results of PLS-SEM," European Business Review, Vol. 31 No. 1, pp. 2-24. https://doi.org/10.1108/EBR-11-2018-0203

Hair, J. F., Ringle, C. M., & Sarstedt, M. (2011). PLS-SEM: Indeed, a silver bullet. Journal of Marketing Theory and Practice19(2), 139-152.

Resolved.

Round 2

Reviewer 1 Report

General Comments:

The authors explore the process of normalization of Social Distancing (S.D.) in the professional lives of healthcare workers. The authors adopted the Normalization Process Theory (NPT) and selected one district in Pakistan that was in the red zone regarding Covid-19 infections to collect data. Data analysis was performed through the Smart-PLS software by following a step-wise process. By explore the effect of four constructs (e.g., coherence, cognitive participation, collective action, and reflexive monitoring) on the normalization of S.D. They find that weak coherence and cognitive participation in the normalization of S.D. They further conclude that the work to improve coherence and cognitive participation may result in stronger relationships among the latent variables and the normalization of S.D.

However, the analysis does not reach a satisfactory level. I do have several major concerns as below.

Major Concerns

1.       In the Introduction, the practical and theoretical motivation for Social Distancing is weak.  From the practice, the authors just describe the S.D. is a part of Nonpharmaceutical Interventions in line 39; from the theory, authors refer to Normalization Process Theory but without any explanation, which is not convincing.  In addition, there's no innovation point description and more specific management implication.

2.       The motivation of the research is weak. 

3.       The introduction part lacks a description of innovation and research value, so I think the motivation of this problem is very weak.

Author Response

Thanks for the constructive comments. In order to incorporate the suggestions, we have redesigned the introduction and the implications part of the study, where we try to give the motivation of SD by explaining, why SD is more important than other nonpharmaceutical interventions in the context of the low-middle-income countries and specifically in Pakistan. We illustrate the reason for selecting NPT as a framework for this study. We describe the innovation and value of the study. We also redesign the implications of the study by including the practical and theoretical implications.

So, we believe that we address all the valuable comments of the respected reviewers.

The updated parts are given below:

Introduction: This study explores the normalization of Social Distancing (SD) in the professional lives of healthcare workers. The COVID-19 pandemic has strained the global healthcare systems and caused extreme worldwide socioeconomic disorders such as deferment and termination of educational, political, cultural, and religious events (Sarwar et al., 2020), ultimately has opened a new pandora's-box. The overall situation of the pandemic motivates the policy institutions to provide scientific support. So, In the absence of Pharmaceutical Interventions, The World Health Organization (WHO) provides policy guidelines for implementing Nonpharmaceutical Interventions (NPIs). These include (but are not limited to) contact tracing, quarantine, and SD. 

While explaining the importance of SD for Low-Middle Income Countries (LMICs), Guimaraes, et al. (2021) demonstrated that other NPIs would increase the financial consequences. In Pakistan, the doctor-to-patient ratio in 2019 was 1.1 doctors per one thousand patients. The ratio is continuously growing. It was 0.93 in 2015, 0.96 in 2016, 1 in 2017, and 0.98 in 2018. (Organization, WH 2019). However, it is considered the least number of human resources for health and below the WHO standard (Khan, 2019). So, the lack of human resources for health in a country with a high population made implementing social measures difficult, specifically for healthcare workers in their professional lives. So, it is pertinent for Pakistan to understand the implementation of NPIs and, more specifically, the social measures – SD. Moreover, healthcare workers must follow SD because they are more prone to the disease because of their job nature. Razaq et al. (2020) observed that ten percent of the reported infection are among healthcare workers. If healthcare workers did not follow the SD, they could be the disease carriers. So, considering the importance of SD in the context of healthcare workers' professional lives, we take SD as a core social measure in this study.

The implementation scholars focused on various aspects of the implementation and explored its context, enablers, challenges, innovation (Rostami, Ashcroft, and Tully, 2018), and multi-actor perspectives (Webster, Ekers, and Graham, 2016). While studying the pandemic, Lotta, Coelho, and Brage (2020) observed the vital role of frontline workers. So, to comprehend the implementation context, it is pertinent to involve the policy actors. Anthony (2021) concluded that it is significant for healthcare workers to observe SD. So, it is pertinent to explore the normalization of SD in the lives of healthcare workers in the local contexts so that developing countries can make better policies in the case of any future pandemic. So, our study focuses on the core research question: What is the normalization process of SD in the professional lives of healthcare workers?

The implementation of healthcare interventions can be disrupted in multiple ways: patient examination protocols, the nature of the disease, the patient-doctor relation, and the broader policy context. Consequently, there is a need to attend to how the social measures are implemented by the policy institutions and adopted by the target population. This study focuses on implementing a complex healthcare intervention in a more focused and micro context, i.e., in the professional lives of healthcare workers. We will describe how the relevant policy actors, specifically the healthcare workers made sense of the intervention, engaged with it, what resources are required, and how they can appraise it. At the same time, our study is focused on implementing a specific social intervention: SD, in clinical settings. The healthcare intervention is considered a complex intervention. Over the last few decades, many conceptual frameworks, models, and theories have been proposed, tested, and recommended to understand and explore complex interventions (Greenhalgh et al., 2017: Damschroder et al., 2009), making the study of healthcare interventions as an interesting and challenging domain. The understanding of the overall process of implementing SD used a robust methodological design, and here we analyze the normalization process of the intervention.

For this study, the authors adopted the Normalization Process Theory (NPT), extensively discussed in the implementation literature, explaining the implementation process of complex and innovative healthcare interventions (Morrison and Mair, 2011). The Implementation of SD in managing Covid-19 is a unique and complex healthcare intervention. So, NPT support as a theoretical framework for understanding the implementation/normalization of SD in the professional lives of healthcare workers. For data collection, we selected one district in Pakistan that was in the red zone regarding Covid-19 infections. Data analysis was performed through the Smart-PLS software by following a step-wise process. Resultantly, the study results explain the relationships of the independent variables with the implementation of SD in the focused and micro context, i.e., in the professional lives of healthcare workers.

The complex environment of the Covid-19 pandemic, the importance of SD for LMICS, and the policy actors/healthcare workers emanate the need for the study. The study fulfills the methodological aspect of theory application and quantitative design and explores the contemporary implementation dilemma. The study addresses the translational gaps: know-do, policy research practice, and think-do. The study findings help the LMICs to successfully implement complex healthcare interventions in the current and future healthcare crises and can make the frontline workers/healthcare workers more oriented towards implementing healthcare interventions. The study satisfies the current implementation paradigms and will intensify the implementation, pandemic, and Structural Equation Modelling (SEM) literature.

The next part explains the objectives and needs of the study. In the literature part, the authors explored the previous research on the normalization of healthcare interventions. The next part presents the theoretical model, constructs, and hypothesis. The methodology part consists of the research design, ethical statement, study area, research instrument, sample design, data coding, data collection, testing of the common source bias, ANOVA-based tests (assumptions of normality and non-parametric analysis), and finally, the application of SEM with its reasoning leading towards the results and conclusion of the study.

Implications: Our study addresses the management/policy and theoretical implications. For the practical, policy, or management implication, our study addresses the translational gaps: know-do, policy research practice, and think-do.

1.     There exist well-known and subtle gaps in translating research results into policy and practice (Eccles et al., 2009), such as the know-do gap: the development and implementation of innovative healthcare interventions in practice with the intended users (Woolf, 2008) such as the healthcare workers in our case. Our study identifies the loopholes in the implementation process by understanding the normalization of an innovative healthcare intervention, so it addresses the know-do gap.

2.     Another well-known gap relates to using health research results to inform health policy (Mackenzie et al., 2010). In our case, the study results represent the weak coherence (sense-making) and cognitive participation (engagement) on the part of healthcare workers that would inform the policy institutions to put more effort into sense-making and engagement of the healthcare workers in Pakistan. This can lead to stronger relationships among the latent variables and the implementation-normalization of a healthcare intervention.

3.     However, the think-do gap illustrates that the groups or individuals (healthcare workers in our case) involved in the implementation should consider the nature and complexity of the implementation work to better inform their actions. Our study explores the implementation of a complex healthcare intervention through what resources they need to enact it and how they can appraise it. The resources required and the appraisal's tendency depend upon the intervention's nature and complexity. So, our study also informs about the think-do gap while understanding the normalization of a complex healthcare intervention in the healthcare workers' professional role.

The implementation scholars believe that the greater and more rational utilization of the theoretical approaches can address the translational gaps (Davies, Walker, and Grimshaw, 2010: Eccles, Weijer, and Mittman, 2011) and can develop, enhance and verify the information about the theories (de Brún et al., 2016). As for the theoretical or methodological part, our study contributes to the theory application and quantitative design and explores the contemporary implementation dilemma. In our case, we apply NPT to understand the implementation of healthcare intervention, and the results can positively contribute to managing any future pandemic/healthcare crisis. So, LMICs can learn from this research conducted in the local context of Pakistan. Resultantly, policy institutions can make better policies, and the target audience can better understand the loopholes in the implementation process. The study tests and accepts the four hypotheses, resulting in the applicability of NPT (application of theory) and the SEM (quantitative design) in the case of exploring the Covid-19 DP&C policy and specifically the normalization of SD.

Reviewer 2 Report

The author has incorporated all the required changes so the manuscript can be accepted for publication. 

Author Response

Thanks for your comments. We believe that your comments/suggestions helped us improve the quality of the manuscript per the academic and journal standards.